# The Analysis of the Human Megakaryocyte and Platelet Coding Transcriptome in Healthy and Diseased Subjects

**DOI:** 10.3390/ijms23147647

**Published:** 2022-07-11

**Authors:** Koenraad De Wispelaere, Kathleen Freson

**Affiliations:** Department of Cardiovascular Sciences, Center for Molecular and Vascular Biology, University of Leuven, 3000 Leuven, Belgium; koen.dewispelaere@kuleuven.be

**Keywords:** platelets, megakaryocytes, transcriptomics, rare disease bioinformatics, inherited platelet disorders, single-cell RNA sequencing, bulk RNA sequencing, megakaryopoiesis, thrombopoiesis

## Abstract

Platelets are generated and released into the bloodstream from their precursor cells, megakaryocytes that reside in the bone marrow. Though platelets have no nucleus or DNA, they contain a full transcriptome that, during platelet formation, is transported from the megakaryocyte to the platelet. It has been described that transcripts in platelets can be translated into proteins that influence platelet response. The platelet transcriptome is highly dynamic and has been extensively studied using microarrays and, more recently, RNA sequencing (RNA-seq) in relation to diverse conditions (inflammation, obesity, cancer, pathogens and others). In this review, we focus on bulk and single-cell RNA-seq studies that have aimed to characterize the coding transcriptome of healthy megakaryocytes and platelets in humans. It has been noted that bulk RNA-seq has limitations when studying in vitro-generated megakaryocyte cultures that are highly heterogeneous, while single-cell RNA-seq has not yet been applied to platelets due to their very limited RNA content. Next, we illustrate how these methods can be applied in the field of inherited platelet disorders for gene discovery and for unraveling novel disease mechanisms using RNA from platelets and megakaryocytes and rare disease bioinformatics. Next, future perspectives are discussed on how this field of coding transcriptomics can be integrated with other next-generation technologies to decipher unexplained inherited platelet disorders in a multiomics approach.

## 1. Introduction to Megakaryopoiesis and Platelet Formation with a Focus on RNA

Megakaryocytes (MKs) are large polyploidic blood cells that originate in the bone marrow and, in their mature state, release platelets into the blood circulation. MKs differentiate from hematopoietic stem cells (HSCs) located in the bone marrow. The classical differentiation model for megakaryopoiesis states that HSCs divide asymmetrically into multipotent progenitor cells, which lose their self-renewal capacity and multipotency over time. This leads to bipotent megakaryocytic/erythroid progenitor (MEP) cells, which, as the name suggests, can differentiate into MKs or erythrocytes. In the former case, they differentiate into unipotent MK progenitor cells and then MK precursor cells [1,2]. Immature MKs can measure up to 30 µm with a high nucleus-to-cytoplasm ratio, while, in a mature state, they can measure up to 160 µm in extremis. During maturation, MKs also become polyploid through endomitosis by repeated DNA replication without cell division and can contain up to 128N chromosomes, with the majority of MKs having 16N. The large amount of DNA in each cell contributes to a high concentration of RNA at the moment that platelets are formed by pinching off cytoplasmic protrusions, called proplatelets. During this process, proplatelets receive RNA and proteins from their parent MK cells (Figure 1) [3,4].

Most studies use in vitro-cultured MKs because of the low yield and difficulty of extracting bone marrow-derived MKs. In vitro MKs can be procured by differentiating induced pluripotent stem cells (iPSCs) through sequential cytokine cocktails (directed differentiation) [6,7,8,9] or through overexpression of key transcription factors (forward programming) [10,11]. The most widely used method for in vitro MK production starts from CD34+ HSCs after adding cytokines [12]. Nakamura et al. [13] generated a stable immortalized MK cell line (imMKCL) from iPSCs. Although it is well known that MKs are the precursor cells of platelets [14], the exact mechanism of differentiation has long been disputed. The most notable theories are the ‘explosive fragmentation theory’, which hypothesizes that MKs release mature platelets by cell apoptosis [15], and the ‘proplatelet theory’, which suggests that MKs form numerous long-branching cytoplasmic protrusions, called proplatelets, with a volume several hundred times that of platelets. These proplatelets then detach, remodel and fragmentate into individual platelets, mainly by the shear stress in the bloodstream [16,17,18,19,20]. The advent of intravital microscopy and observations of the bone marrow in real time have set the consensus quite firmly onto the ‘proplatelet theory’, complemented by the confirmation that mature MKs extend their protrusions through the sinusoid vessel barrier when releasing proplatelets. The proplatelets are rapidly swept away by the blood circulation and remodeled further downstream in the circulation before becoming bona fide platelets [18,21,22,23,24]. Though the ‘explosive fragmentation theory’ is unlikely to occur during homeostasis, it could happen under pathological conditions, such as during the release of large amounts of platelets in response to inflammatory stimuli such as IL-1a [25].

Platelets are the second most prevalent cell in the blood and play a pivotal role in cardiovascular diseases by maintaining a balance between hemostasis and blood clot formation. Platelets are small (2–4 µm in diameter), anuclear cells that circulate in the blood for 7–10 days in humans, after which they are eliminated in the spleen and liver [26]. In 1947, the presence of RNA in platelets was first reported using a chemical method [27]. The discovery that active protein synthesis in platelets is not dependent on transcription but enabled through mRNA transferred through MKs was first reported by Warshaw et al. [28]. Platelets do exhibit a complex and dynamic transcriptome that is inherited from MKs. This transcriptome includes messenger RNA, ribosomal RNA, transfer RNA and regulatory RNAs such as miRNA [29,30,31]. miRNAs regulate RNA stability and protein translation by binding near complementary sequences in the 3′ UTR of transcripts, and they are highly abundant in platelets and MKs. Their functions as biomarkers and functional modifiers in platelets and MKs have been reported and reviewed by others [32,33,34,35,36,37,38,39]. Our review focuses on mRNA transcriptomes.

Platelets have on average 2.2 fg of RNA present in total, but this amount is 20–40-fold higher in young platelets and decreases with time [40]. Platelets do also exhibit de novo protein synthesis, and the transcriptome dynamically changes in response to inflammatory signals, invading pathogens, cancer or other stressors [4]. In nucleated cells, mRNA has a median lifespan of 7.6–9 h, varying considerably between transcripts [41,42]. Platelet mRNA has a higher stability for reasons not yet completely understood, despite the fact that all typical stability and decay-mediating RNA-binding proteins are present [43,44,45]. It has also been observed that the average length and thermodynamic stability of the untranslated regions of transcripts in platelets are significantly greater than in other cell types [46]. Alternative splicing, in which introns can be (partially) retained and exons can be (partially) removed, strongly diversifies the transcriptome, and this is a process that is proven to be active in platelets [47,48,49,50,51,52]. A higher level of unspliced pre-mRNA, coming from the MKs, in younger platelets and the presence of the spliceosome suggest that this mechanism might serve as a gatekeeper for pre-mRNA processing and protein translation in platelet function [47,53,54,55,56].

Here, we review bulk and single-cell RNA-seq studies that have used MKs and platelets to understand their differentiation processes and their function in normal and pathological conditions. Unlike microarrays, RNA-seq does not require transcript-specific probes and can therefore detect novel, rare and unknown transcripts, non-coding transcripts, alternative splicing, gene fusions, single nucleotide variants and indels.

## 2. Bulk Transcriptomics of Healthy Megakaryocytes

The study of healthy MKs enables the identification of genetic pathways regulating MK differentiation and thrombopoiesis. Given the dynamic nature of MK cultures, first being differentiated from HSCs and later producing platelets themselves, cell sorting is often used to isolate specific MK subpopulations representing different differentiation stages before applying bulk RNA-seq. Different cell-specific markers have been identified, such as CD41 or CD61, markers associated with early MK and erythroid progenitors, and CD42, a glycoprotein expressed during the later stage of MK differentiation [57,58] (different cell markers are shown in Figure 1). For mature MKs, a large sorting nozzle and low pressure are important for maintaining cell viability [59].

The BLUEPRINT (https://www.blueprint-epigenome.eu/, accessed on 5 June 2022) consortium has used bulk RNA-seq extensively in a broad effort to dissect the molecular traits that govern blood cell differentiation in general, thus also including megakaryopoiesis [60]. Cecchetti et al. [61] used bulk RNA-seq on CD34+-derived MKs after selection with CD61+ immunomagnetic beads and leukocyte-depleted platelets to study mRNA for matrix metalloproteinases (MMPs) and their inhibitors (TIMPs). They found that MKs differentially express mRNAs for MMPs and TIMPs, and only a subset of these are transferred to platelets. Human iPSC-derived MKs [62], CD34+ HSC-derived MKs [63] and imMKCL [64,65] have all been used for bulk RNA-seq studies, while similar studies have not yet been performed for primary MKs because of their low yield and the difficulty extracting them from the bone marrow. A comparison between the different types of in vitro MKs using RNA-seq datasets has not been reported. Regarding data analysis, many different bulk RNA-seq pipelines exist and have been reviewed elsewhere [66,67,68,69].

## 3. Bulk Transcriptomics of Healthy Platelets

A critical step for platelet RNA-seq is the sample purity of platelets after their isolation from peripheral blood. A leukocyte depletion step is critical to exclude contamination by leukocytes and their derived vesicles. Therefore, magnetic antibody-coupled beads are mostly used for leukocyte (CD45+) depletion, and afterwards, the purity of the platelet population for RNA-seq should be confirmed by flow cytometry using platelet- and leukocyte-specific markers [57]. RNA sequencing of healthy platelets provides a sequence-level view of their transcriptome and is useful for understanding platelet function, as well as megakaryopoiesis, since the transcriptome of platelets is shaped by genetics and environmental signals from MKs. The human platelet transcriptome has been characterized by bulk RNA-seq as diverse, containing many coding and non-coding transcripts. Londin et al. [70] reported an average of ~9000 unique protein-coding transcripts and ~800 miRNAs. These two types only accounted for half of all reads, which points to the presence of large amounts of other non-coding RNA. Despite the anucleate nature of platelets, non-coding transcripts include abundant miRNAs, retrotransposons, and long and short intronic transcripts [29,71]. Mitochondrially expressed genes also comprise a substantial fraction of the platelet transcriptome, and high transcript levels are present for protein-coding genes related to cytoskeleton function, chemokine signaling, cell adhesion, aggregation and receptor interaction between cells [72]. In a large-scale platelet RNA-seq study of 204 healthy individuals, high expression of *B2M*, *PPBP*, *TMSB4X*, *ACTB*, *FTL*, *CLU*, *PF4*, *F13A1*, *GNAS*, *SPARC*, *PTMA*, *TAGLN2*, *OAZ1* and *OST4* was observed, with substantial consistency between individuals. Additionally, platelets in males have higher *CSF3R* expression, and in older individuals, *KSR1* is upregulated [73]. Another large-scale platelet RNA-seq study in 290 healthy subjects was performed to evaluate gene expression in platelets and iPSC-derived MKs from the same subjects, showing a high overlap of 91.3% between these transcriptomes [62]. The platelet transcriptome correlates strongly between individuals but associates weakly with the platelet proteome [70]. Platelets have been proven to inherit an active spliceosome from their parent MKs, diversifying the transcriptome and proteome [49,54]. Splicing has also been proven, using RNA-seq, to be induced by platelet activation. This modulates protein expression and forms a crucial part of the platelet activation cascade [30]. Interestingly, in the absence of disease, platelet gene expression and splicing are remarkably stable within individuals over time, as observed in a 4-year-long longitudinal RNA-seq study [74]. As with MKs, platelet populations can be subdivided using flow cytometry. Flow cytometry can separate platelets into their smallest and largest 10% subpopulations, providing transcriptomic proof of different functionalities between platelets, with large platelets having transcriptomes more linked to hemostasis and wound healing and the smallest being more associated with vascular cell function [75]. Hille et al. [76] sorted platelets based on age, which they analyzed for ultrastructural, functional and transcriptional differences. This revealed structural and molecular differences between younger, reticulated platelets and older, non-reticulated platelets, explaining the hyperreactivity of the former class. RNA-seq demonstrated the upregulation of transcripts allocated to shape change, aggregation and degranulation in younger platelets.

## 4. Single-Cell Transcriptomics of Healthy Megakaryocytes

For single-cell RNA sequencing (scRNA-seq) methods, the RNA content of each cell receives a unique barcode enabling cell-wise demultiplexing after sequencing. First, microwells were used to isolate single cells in plate-based assays such as smart-seq [77,78]. Newer techniques then transpired through the implementation of technologies such as integrated fluidic circuits (Fluidigm C1 platform). After 2015, this technique was largely overtaken by the advent of droplet microfluidics, in which each single cell is encapsulated into a droplet containing a bead with bead-unique barcodes. Examples of this technique are Hydrop [79] and the Chromium platform from 10X Genomics. Lastly, there are also techniques based on the principle of combinatorial indexing for barcode attachment [80,81,82]. A more thorough review and benchmarking of the many existing scRNA-seq methods have been reported by others [83,84]. These four types of scRNA-seq assays have all been used for studying MKs and their precursors in humans. Different dedicated data analysis pipelines for scRNA-seq data exist and have also been reviewed elsewhere [85,86,87,88]. For analyzing human MKs, plate-based methods (mainly smart-seq2) have often been used [89,90,91]. Sun et al. [91] raised the concern that droplet microfluidic techniques might be unsuitable for scRNA-seq of large, rare and fragile polyploidic MKs, and therefore, they used the plate-based smart-seq2 protocol. The theoretical upper limit cell diameter for 10X Chromium, for example, is 65 µm (width of the capillaries), with an advised maximum of 30 µm. In this case, plate-based or combinatorial indexing methods form a good alternative. Other studies [92,93,94,95] have, however, successfully constructed high-quality MK libraries using 10X Chromium. Table 1 compares the advantages of the two most frequently used scRNA-seq methods, plate-based and droplet-based. The number of single-cell transcriptomic studies focusing on megakaryopoiesis and MKs is rapidly increasing but is still limited. These studies are summarized in Table 2 and are briefly discussed below.

In MKs and their precursors, scRNA-seq has been proven useful for studying their dynamic transcriptome throughout their various developmental trajectories [89,90,91,94,96]. Polyploidization during maturation, for example, is associated with a shift from transcripts associated with platelet degranulation, coagulation, hemostasis, wound healing and vesicle-mediated transport towards transcripts associated with translational initiation, elongation and termination, protein localization and cellular protein complex disassembly [89]. The heterogeneity within a single type of MK cell or their precursors, as seen in many scRNA-seq studies [90,91,92,97], would be missed by bulk RNA-seq methods. Notably, a subset of HSCs has often been described that is biased towards the myeloid lineage and is transcriptionally primed to directly differentiate into MK progenitor cells [58,89,91,92,95,97,98]. Half of all MKs are thought to differentiate through this mechanism, which might also play a role in the rapid replenishment of platelets in response to acute demands in pathological conditions, such as in the wake of a myocardial infarction [89] or in myelofibrosis [95,97]. *THBS1* has been identified as an early marker for this CD14+ MK-biased HSC subpopulation [92]. In Sun et al. [91], three MK subpopulations were reported with distinct transcriptomes linked to the inflammatory response, HSC niche interactions and platelet generation. scRNA-seq has also driven transcriptomic research into primary MK cells [89,94], for which extensive sample purification is needed, most often through flow cytometry [90,99]. The study of Lu et al. [90] identified key genes involved in MEP fate determination and demonstrated that differential modulation of cell cycle speed dictates MEP differentiation into either erythrocytes or megakaryocytes using scRNA-seq.

Although scRNA-seq analysis has been carried out on MK differentiation in vivo [89,90,91] and in vitro from iPSCs or CD34+ HSCs [94], the transcriptomic difference between different types of in vitro and in vivo MKs is largely unknown. Recently, the first single-cell study on this subject [94] assessed the transcriptomic differences between megakaryopoiesis in vivo and in vitro through iPSC forward programming. Cells from the latter type were compared to primary hematopoietic stem and progenitor cells from the bone marrow, and they found that the in vitro cells did not pass through states resembling primary HSCs or MPP cells. In a further stage of differentiation, the cell cultures did, however, go through an MK progenitor stage, with a very similar transcriptional signature to their in vivo counterparts. Additional research should be carried out to decipher transcriptional differences between in vivo and different types of in vitro-cultured MKs to confirm the utility of the latter for studying defects in megakaryopoiesis.

**Table 2 ijms-23-07647-t002:** scRNA-seq research in megakaryocytes and precursors in humans.

Reference	Cell Type(S)	Summary of Results	Technology
Choudry et al. (2021) [73]	MKs and HSCs	MKs in lower ploidy states highly express platelet-specific genes. As polyploidization increases and the cell prepares for thrombopoiesis, gene expression is redirected towards transcriptional programs involved in translation and posttranslational processing. Two MK-biased HSC subpopulations were also observed and shown to originate from the BM. Finally, BM MKs from patients with recent myocardial infarction showed a specific gene expression signature that supports the modulation of MK differentiation in this thrombotic state.	G&T-seq
Estevez et al. (2021) [98]	HSCs	The effect of germline monoallelic mutations in RUNX1, found in patients suffering from familial platelet disorder with a predisposition to myeloid malignancy, was studied by inserting the patient mutations into iPSC-derived hematopoietic progenitor cells (iHSCs) and performing scRNA-seq. There was found to be a marked deficiency of MK-biased iHSCs in mutated cultures, and gene sets that were upregulated included response to stress, regulation of signal transduction and immune signaling-related gene sets. An increased sensitivity to transforming growth factor β1 and an increase in the stress pathway through upregulation of c-jun N-terminal kinase-2 phosphorylation were observed.	10X Chromium
Lawrence et al. (2022) [94]	In vitro differentiating cells from iPSCs and HSCs up to MKs	Analysis of iPSC-derived MK differentiation and transcriptomic comparison with primary hematopoietic stem and progenitor cells. The in vitro cells do not pass through states resembling HSCs or MPPs as seen in vivo, but the further differentiated MK progenitor cells do exhibit a very similar transcriptome to their in vivo counterparts. A surface marker panel is described for MK progenitors, allowing for selection from culture and for insights into this intermediary state.	10X Chromium and smart-seq
Liu et al. (2021) [90]	MKs	Cellular heterogeneity within MKs was mapped, and an MK subpopulation with high enrichment of immune-associated genes was identified. The immune signature could be traced back to the progenitor stage, and two surface markers, CD148 and CD48, were identified. This type of MK can respond rapidly to immune stimuli both in vitro and in vivo, exhibiting high expression of immune receptors and mediators, which might act as immune-surveillance cells.	Smart-seq
Lu et al. (2018) [96]	MEPs, common myeloid progenitors, and MK and erythroid progenitors	MEPs have a distinct gene expression signature that represents a continuous transition state from common myeloid progenitor cells to MK and erythroid progenitor cells.	Fluidigm C1
Psaila et al. (2017) [95]	CD34+ peripheral blood cells	Myelofibrosis causes an increased number of immature/low ploidy MKs with an altered transcriptome. Patient HSPCs have increased expression of MK-associated genes, including VWF and ITGA2B. Patient CD34+ progenitor cells showed increased expression of PF4 and TGFβ.	10X Chromium
Psaila et al. (2020) [97]	HSPCs	MK-biased hematopoiesis in myelofibrosis was observed, with heterogeneous MKp showing a highly expressed fibrosis signature and an aberrant metabolic and inflammatory signature. Targeting the aberrant expression of surface G6B may selectively ablate the myelofibrosis HSPC clone.	10X Chromium
Riemondy et al. (2019) [93]	Lymphocytes and MK mixture	The introduction of a method for resampling cell-type-wise, cell-wise or sample-wise from an existing complex scRNA library.	10X Chromium
Sun et al. (2021) [91]	Human and mouse MKs	Three distinct MK subpopulations were observed to possess gene signatures related to platelet generation, HSC niche interaction or inflammatory response. The first type of MK was mostly found near blood vessels, and the second was near HSCs. The third type, containing a gene signature related to the inflammatory response, was lower in ploidy, consisted of 5% of MKs and was capable of engulfing and digesting bacteria and stimulating T cells in vitro.	Smart-seq
Wang et al. (2021) [92]	Human MKs	A comprehensive single-cell transcriptomic landscape of human MKs was constructed where MKs show cellular heterogeneity with distinct metabolic and cell cycle signatures. CD14+ MKs with immune characteristics were generated along a distinct trajectory, and THBS1 was identified as an early marker for MK-biased endothelial cells from human embryonic stem cells.	10X Chromium

## 5. Single-Cell Transcriptomics of Healthy Platelets

Single-cell RNA-seq has not yet been used for platelets because of their extremely low RNA content (~2.2 fg/platelet). The leading edge of scRNA-seq assays might enable this in the future, with single-cell libraries currently being created from cDNA contents of 1–2 ng/cell by the much-used commercial 10X Chromium platform. Alternatively, the Smart-seq v4 Ultra Low Input RNA kit has been validated to an RNA concentration of 2 pg of total RNA per cell by the manufacturer. An scRNA-seq protocol called ultra-low RNA sequencing (ulRNAseq) was optimized for RNA concentrations as low as 0.5 pg per cell [100]. scRNA-seq of platelets would lead to novel insights into platelet heterogeneity, which has been proven to exist using proteomic and transcriptomic techniques other than scRNA-seq, such as platelet contraction cytometry, immunohistochemistry, epifluorescence microscopy or microarrays, and is reviewed in [101].

## 6. Platelet and Megakaryocyte Transcriptome to Decipher Inherited Platelet Disorders

Changes in the MK and platelet transcriptome have been studied in diverse pathological conditions, as recently reviewed by Davizon-Castillo et al. [5], who focused on the changed transcriptomes under acquired conditions such as sepsis, myocardial infarction and viral infection and during cancer, sickle cell disease and lupus erythematosus. In this review, we focus on monogenic conditions as the cause of changes in the MK and platelet transcriptome, as RNA-seq studies were also performed for inherited platelet disorders (IPDs). IPDs are an extremely heterogeneous group of diseases, affecting both platelet formation and their function. Patients with a defect in platelet formation present with abnormally low platelet counts, named thrombocytopenia. The clinical phenotypes of IPD patients cover a wide spectrum, ranging from mucocutaneous bleeding diathesis (epistaxis, gum bleeding, purpura and menorrhagia) to multisystemic disorders and malignancies. Consequently, the impacts of these disorders also range from almost negligible to life threatening [102,103,104]. MK and platelet transcriptomics can be used to assist gene discovery processes of unexplained IPDs and to provide novel insights into the disease mechanism of known IPDs. Examples of both are discussed below.

The first RNA-seq-enabled discovery for an IPD was the identification of *NBEAL2* variants as the cause of gray platelet syndrome using platelet RNA-seq [105]. Abnormal reads were detected, showing intron retention in *NBEAL2* for platelets from a gray platelet syndrome patient due to a splice variant. A more recent bulk RNA study for gray platelet syndrome reported widespread differences in platelet, neutrophil, monocyte and CD4 lymphocyte transcriptomes between patients and controls, but loss of function of *NBEAL2* does not seem to affect the transcriptional output of α-granule-associated genes in platelets [106]. Variants in *ETV6*, a transcription factor that plays a central role in hematopoiesis and malignant transformation, have been implicated in thrombocytopenia [107]. Platelet RNA-seq for ETV6-deficient patients showed decreased expression of platelet-specific transcripts, including reduced levels of several cytoskeletal transcripts. Heremans et al. [108] performed bulk RNA-seq of CD34+ HSC-derived MKs from Roifman syndrome patients with genetic variants in *RNU4ATAC*, which regulates minor intron splicing. Significant minor intron retention was detected in the patient transcriptomes for 354 MK genes, including many linked to the thrombocytopenia phenotype of platelets. Lentiviral-transduced CD34+ HSCs were generated for wild-type SRC and E527K-SRC and differentiated into MKs before applying bulk RNA-seq [63]. The E527K SRC variant causes thrombocytopenia, and MK transcriptomes for this variant showed 852 significant differentially expressed genes, with interferon I signaling as the most interesting pathway after confirmation with MK proteomics. The effect of germline monoallelic variants in *RUNX1*, found in patients suffering from familial platelet disorder with a predisposition to myeloid malignancy, was studied by inserting the patient variants into iPSC-derived HSCs and performing scRNA-seq [98]. There was found to be a marked deficiency of MK-biased HSCs in mutated cultures, and gene sets that were upregulated included the response to stress, regulation of signal transduction, and immune signaling-related gene sets, among others. In Lentaigne et al. [57], variants in the transcription factor *IKZF5* were found to be causal for thrombocytopenia. Comparison of platelet, monocyte, neutrophil and CD4+ T cell bulk RNA-seq data from patients and controls showed 1194 differentially expressed genes in platelets, while only 4 genes were differentially expressed in each of the other cell types. The pathways downregulated in platelets involved platelet function, hemostasis and membrane- and vesicle-mediated transport.

The challenge ahead will be to now use platelet and MK RNA-seq datasets to discover new IPD genes, as, to date, most transcriptome studies have only endeavored to decipher defective platelets or MKs for known IPDs. Instead of using RNA-seq data for gene expression and pathway analyses, additional information can be retrieved, such as differences in splicing and monoallelic expression. More recently, the advent of specialized data analysis methods has been directed at finding and statistically assessing outliers in transcriptomic data, and this has stimulated progress in the field of transcriptomics in rare disease research. Outlier detection tools for gene expression (e.g., OUTRIDER) [109], aberrant splicing (e.g., FRASER) [110] and monoallelic expression have been developed, which are all combined in the DROP pipeline [111]. In these methods, confounding factors are corrected by using the machine learning method of a denoising autoencoder. Its future application is discussed in the final section.

## 7. Conclusions and Future Perspectives

It would be very interesting to compare the transcriptomes of platelets, primary MKs and in vitro-generated MKs from single donors using a similar RNA-seq technique. Ideally, since MK populations are very heterogeneous at different differentiation stages, sc-RNA-seq would be best, but this method is still not able to generate transcriptomes for platelets due to their low RNA content. As these methods improve, it is expected that this will become possible in the future. Such a comparative gene expression experiment would improve our current understanding of MK differentiation, thrombopoiesis and platelet function under healthy conditions. It will not be possible to generate such datasets for most patients with IPDs, as primary MKs are typically not available.

Another challenge is the interpretation and functional validation of RNA-seq results. Most studies focus on the detection of differentially expressed genes and pathways. However, it remains very difficult to understand the biological relevance of these changes. One can focus on a subset of the most significant up- or downregulated genes and validate these changes using qRT-PCR or immunoblot analysis, but this is of course a biased approach. It is also possible to validate a pathway using functional MK or platelet studies that have been designed to study the pathway of interest. The main issue remains that most studies are biased by highly expressed genes, while genes with weak expression but still important functions in MKs and platelets are typically missed. Better analytical approaches and larger datasets are needed. At least some of the challenges of associating IPDs with rare variants, in both coding and non-coding regions of the genome, can be overcome by the integration of multiple layers of information from different assay types. These different assays can be performed simultaneously in single cells by techniques such as G&T-seq [112] (gDNA and transcriptome), 10X Multiome or SHARE-seq [113] (transcriptome and chromatin accessibility) or TEA-seq [114] (transcriptome, chromatin accessibility and cell surface epitope detection). This multiomics approach is increasingly being used in the field of cardiovascular disease [115], although not yet at the single-cell level. By using chromatin accessibility and histone modification data specific for MKs combined with WGS data, the deletion of an important regulatory element near *HDAC6/GATA1* was detected as the cause of thrombocytopenia and autism [116]. It was also hypothesized that a combination of platelet RNA-seq and WGS data would be of use to identify which variants in regulatory elements could be causal for an IPD [117]. Combining platelet RNA-seq and WGS data from individuals has led to important new discoveries in the study of platelet disorders [57,63,105,106,107,108], but its application in rare disease bioinformatics, as mentioned above, promises important novel discoveries in rare types of IPDs. Changes in gene expression and aberrant splicing events detected in the platelet transcriptome can be explained by variants in both coding and non-coding parts of the genome (Figure 2A,B). To further causally link aberrant splicing to variants, a deep learning model such as SpliceAI [118] can be used to estimate the probabilities of mutations causing aberrant splicing (Figure 2B). This 32-layer deep neural network estimates the probability of splicing events as a consequence of mutations in both coding and non-coding parts of the genome based on the change in sequence. Finally, the effect of heterozygous variants found in the WGS data could also be assessed by checking for monoallelic expression (Figure 2C). In a study by Murdock et al. [119], 115 undiagnosed patients with suspected Mendelian conditions underwent sequencing by RNA-seq and WES or WGS using whole blood and skin fibroblasts. The transcriptome-directed approach resulted in an additional diagnostic rate of 12% across the entire cohort directly due to the addition of RNA-seq data, solving 17% of cases that could not be diagnosed using only WES/WGS. This highlights the possible utility of combining these two datasets in the study of IPDs. However, platelet transcriptome data have not yet been generated in a diagnostic setting for IPDs.

## Figures and Tables

**Figure 1 ijms-23-07647-f001:**
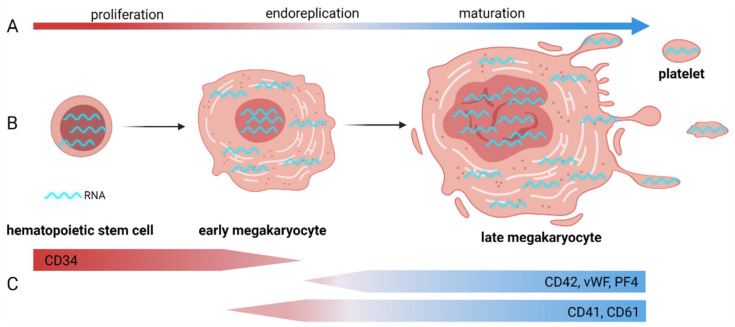
(**A**) Stages in megakaryopoiesis and thrombopoiesis; (**B**) RNA metabolism in maturing megakaryocytes and transfer to platelets during thrombopoiesis; (**C**) different cell markers associated with stages in MK and platelet differentiation (adapted from Davizon-Castillo et al. [5]).

**Figure 2 ijms-23-07647-f002:**
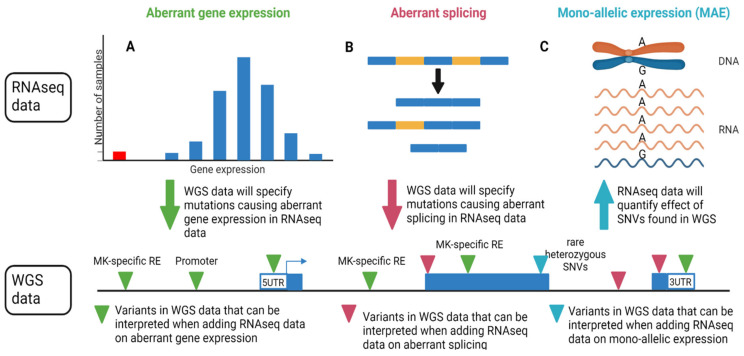
Schematic of information gained by combining RNA-seq data and Whole Genome Sequencing (WGS) data. (**A**) RNA-seq gene expression outliers offer insight into the transcriptomic effect of coding and non-coding patient mutations in WGS data. (**B**) The probability of a change in DNA sequence causing alternative splicing can be predicted by deep learning. Predictions made for patient mutations seen in WGS data can be verified through splicing outlier detection in RNA-seq data. (**C**) The effect of heterogeneous patient mutations from WGS data is also dependent on monoallelic expression, in which one of the two alleles is (partially) unused. This can be quantified using RNA-seq data.

**Table 1 ijms-23-07647-t001:** Comparison between the two most frequently used scRNA-seq methods.

Plate-Based	Droplet-Based
Higher sensitivity	Higher cell throughput
Better for large and fragile cells	Lower labor intensity
Cheaper setup cost	Cheaper cost per cell

## Data Availability

Not applicable.

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
