# Peer review of "The Analysis of the Human Megakaryocyte and Platelet Coding Transcriptome in Healthy and Diseased Subjects"

_ijms, 2022, doi:10.3390/ijms23147647_

Round 1
Reviewer 1 Report
De Wispelaere and Freson wrote a well-structured review that summarises recent bulk RNAseq and single cell RNAseq studies of human platelets and megakaryocytes. I think this work is of great interest and importance to the field and I only have 2 comments:
1. In the introduction, you state that MKs release platelets via fragmentation and cite a rather old paper from 2005. Nishimura et al showed in 2015 for example that this happens due to inflammatory stimuli like IL-1a (PMID: 25963822) Therefore, MK rupture does most likely not happen during homeostasis but during pathological conditions.
2. The authors also mention that the fragmentation of proplatelets is still unknown. However, shear stress is believed by many studies to be simply the main reason for platelet fission from proplatelets, as shear stress also increases thrombopoiesis (and microparticle formation) of MKs (PMIDs: 17885137, 25411426, 21079248, 24948658) This was nicely visualized for example by Lefrancais et al in 2017 (PMID: 28329764)
Author Response
Dear Reviewer 1,
Thank you for your suggestions, you will find the adjustments made in the attachment.
Kind regards

Reviewer 2 Report
De Wispelaere and Freson provide an informative review describing current knowledge on the platelet transcriptome. Separated in bulk and single cell transcriptome data from healthy platelets and healthy megakaryocytes, the current state-of-the-art, general aspects and recent developments are discussed. Specifically, this review manuscript describes new insights from platelet RNA-seq unravelling the cause of rare inherited platelet disorders. Overall, the text structure of the individual paragraphs should be improved. I have some comments, which the authors should address in a revised manuscript version:
Comments:
On page 2, when discussing the various RNAs of platelets, the role of platelet miRNAs should briefly be summarized. Of the 800 identified miRNAs, which are the most abundant? Here a table might be interesting.
How can impurities of RNA preparations be identified in platelet bulk sequencing data. Is contamination with leukocyte microvesicles problematic in platelet preparations for RNA-seq? Please discuss.
How can results from megakaryocyte and platelet RNA-seq analyses, which are descriptive in nature, be exploited in the design of functional studies? Please discuss.
Comparing transcriptomes from megakaryocyte cultures with the transcriptomes of ex vivo isolated megakaryocytes, how different are those transcriptome datasets and how do the authors categorize the use of cultured megakaryocyte transcriptome analyses?
Based on the author’s comparison of the existing transcriptome studies, can they provide an outline and create a scheme on the major pathways that are differentially regulated between megakaryocytes, pro-platelets, and young vs aged platelets? Such an overview will strongly increase the significance of the review.
Minor comments:
On page 4, line 175 that should be 65 µm and line 176 that should be 30 µm and not 30µm.
Author Response
Dear reviewer 2,
Thank you for your suggestions, you will find the adjustments made in attachments.
Kind regards

Reviewer 3 Report
The current review paper by Wispelaere et al discusses single-cell RNA-seq studies dissecting the coding transcriptome of healthy megakaryocytes and platelets in humans. They have also illustrated the importance of platelet and megakaryocyte transcriptome for the identification of inherited platelet disorders. Overall the article is well written and is of considerable importance to the field. Some minor comments are:
1. It would help if a brief section is added to discuss other studies (besides inherited platelet disorders) where the platelet/megakaryocyte transcriptome is altered.
2. A brief table can also be added to discuss the scRNAseq techniques (along with advantages and disadvantages) used to study the MK and their precursors.
Author Response
Dear reviewer 3,
Thank you for your suggestions, you will find the adjustments in the attachment.
Kind regards
